# Pregnant Women Infected with Zika Virus Show Higher Viral Load and Immunoregulatory Cytokines Profile with CXCL10 Increase

**DOI:** 10.3390/v13010080

**Published:** 2021-01-08

**Authors:** Elizabeth Camacho-Zavala, Clara Santacruz-Tinoco, Esteban Muñoz, Rommel Chacón-Salinas, Ma Isabel Salazar-Sanchez, Concepción Grajales, Joaquin González-Ibarra, Victor Hugo Borja-Aburto, Thomas Jaenisch, Cesar R. Gonzalez-Bonilla

**Affiliations:** 1División de Laboratorios de Vigilancia e Investigación Epidemiológica, Instituto Mexicano del Seguro Social, Mexico City 07760, Mexico; lizbethcmz@gmail.com (E.C.-Z.); clara.santacruz@imss.gob.mx (C.S.-T.); eban10@hotmail.com (E.M.); 2Departamento de Inmunología, Escuela Nacional de Ciencias Biológicas, Instituto Politécnico Nacional, Mexico City 11340, Mexico; rommelchacons@yahoo.com.mx (R.C.-S.); isalazarsan@yahoo.com (M.I.S.-S.); 3Coordinación de Control Técnico de Insumos, Instituto Mexicano del Seguro Social, Mexico City 07760, Mexico; con.graja@gmail.com; 4Coordinación de Investigación en Salud, Instituto Mexicano del Seguro Social, Mexico City 06720, Mexico; Joaquin.gonzalezi@imss.gob.mx; 5Dirección de Prestaciones Médicas, Instituto Mexicano del Seguro Social, Mexico City 06720, Mexico; victor.borja@imss.gob.mx; 6Heidelberg Institute of Global Health (HIGH) and Tropical Medicine, Department of Infectious Diseases, Heidelberg University Hospital, 69120 Heidelberg, Germany; thomas.jaenisch@uni-heidelberg.de

**Keywords:** Zika virus infection, pregnant women, cytokines

## Abstract

Background: Zika virus (ZIKV) infection during pregnancy usually shows only mild symptoms and is frequently subclinical. However, it can be vertically transmitted to the fetus, causing microcephaly and other congenital defects. During pregnancy, the immune environment modifications can alter the response to viruses in general and ZIKV in particular. Objective: To describe the role of pregnancy in the systemic pro- and anti-inflammatory response during symptomatic ZIKV infection. Materials and Methods: A multiplex assay was used to measure 25 cytokines, chemokines, and receptors in 110 serum samples from pregnant and nonpregnant women with and without ZIKV infection with and without symptoms. Samples were collected through an epidemiological surveillance system. Results: Samples from pregnant women with ZIKV infection showed a higher viral load but had similar profiles of inflammatory markers as compared with nonpregnant infected women, except for CXCL10 that was higher in infected pregnant women. Notably, the presence of ZIKV in pregnancy favored a regulatory profile by significantly increasing anti-inflammatory cytokines such as interleukin (IL)-10, receptors IL-1RA, and IL-2R, but only those pro-inflammatory cytokines such as IL-6, interferon (IFN)-α, IFN-γ and IL-17 that are essential for the antiviral response. Interestingly, there were no differences between symptomatic and weakly symptomatic ZIKV-infected groups. Conclusion: Our results revealed a systemic anti-inflammatory cytokine and chemokine profile that could participate in the control of the virus. The anti-inflammatory response in pregnant women infected with ZIKA was characterized by high CXCL10, a cytokine that has been correlated with congenital malformations.

## 1. Introduction

Zika virus (ZIKV) is a flavivirus transmitted by the bite of female mosquitoes of the genus *Aedes*, mainly *A. aegypti* and to a lesser degree *A. albopictus.* The virus was first identified in a rhesus macaque in Africa in 1947 and isolated from a human in Nigeria in 1954. ZIKV infection in humans remained a sporadic zoonotic disease in Africa and Southeast Asia until the first large urban outbreak that struck Micronesia in 2007 [1]. Over the next five years, the disease spread to Polynesia, Easter Island, the Cook Islands, and New Caledonia and was introduced and disseminated rapidly in South America in 2015 and to Central America and the Caribbean, including Mexico, the following year [2,3,4]. After these large epidemics, ZIKV infection has remained endemic with a low transmission rate in South and Central America, including Mexico, Peru, and Colombia. The genomic analysis of ZIKV strains isolated between 1968 and 2002 revealed two circulating lineages, African and Asian. Although the rapid spread of the Asian lineage through the Americas was mainly attributed to the movement of people related to tourism and sporting events, phylogenetic and molecular analyses establish the introduction of ZIKV into the Americas in 2013, more than 12 months before it was detected in Brazil. [5] ZIKV infection has a wide variety of clinical presentations and may be asymptomatic, a self-limiting febrile illness, or a severe illness with chronic complications. Recent studies suggest that asymptomatic cases account for less than 30% [6] and not 80%, as previously estimated [7]. The characteristic clinical features include a fine maculopapular skin rash that is diffusely distributed and is associated with two or more of the following symptoms: fever, headache, nonpurulent conjunctivitis, arthralgia, pruritus, and retro-ocular pain [4]. However, the nonspecific clinical symptoms of ZIKV infection are often misdiagnosed as other arboviral infections, especially dengue virus (DENV) infection. Since ZIKV has been detected in many body fluids, including blood, urine, saliva, tears, and semen, where it may persist for several weeks after active infection, it has been suggested that the disease may spread through multiple routes of infection [8].

The two main public health concerns relating to ZIKV infection are its association with neurological manifestations, mainly Guillain–Barré syndrome, meningitis, and meningoencephalitis in the adult population [9] and the vertical transmission during pregnancy that is linked to the development of brain abnormalities in the fetus and other anomalies related to the congenital Zika syndrome [4,10]. The risk factors associated with ZIKV infection in pregnancy and congenital Zika syndrome remain unclear and vary widely by country; however, its incidence was reported to be particularly high in Brazil [2].

ZIKV infection produces pro-inflammatory innate and adaptive immune responses. The innate immune response is mainly mediated by the induction of types I and III IFN, which induce an early autocrine antiviral stage in infected cells [11,12]. The adaptive humoral response induces broad neutralizing and protective antibodies. However, it is not clear whether these antibodies can intensify reinfection through antibody-dependent enhancement (ADE), as described for DENV [13,14]. In fact, antibodies to DENV cross-react with ZIKV but exhibit limited neutralization activity and mediate ADE [13]. ZIKV infection also induces robust CD4+ T cell responses, characterized by the production of type 1 cytokines, IFN-γ, tumor necrosis factor-α (TNF-α), and interleukin (IL)-2, whereas CD8+ T cells contribute to the protective mechanisms against primary ZIKV infection through their specific cytotoxic activity, and upregulation of IFN-γ and TNF-α [15]. The systemic inflammatory response during the acute stage of Zika fever is characterized by high circulating levels of the cytokines and chemokines IL-9, IL-17A, and CXCL10 [16]. Interestingly, a bimodal expression of IL-1β, IL-13, IL-17, TNF-α, and IFN-γ, CXCL8, CCL2, CCL5, and the growth factor granulocyte colony-stimulating factor (G-CSF) has been seen [17]. This strong inflammatory pattern during acute infection is also accompanied by high levels of IL-1RA and IL-4. During late-stage disease and convalescence, CXCL10, IFN-γ, IL-10, CCL4, and CXCL12 are increased, while IL-1RA, CXCL8, and CCL2 are decreased [16], suggesting a functional role for T cells [18]. Viremic patients with moderate symptoms have high levels of CXCL10, CCL2, IL-1RA, CXCL8, and placental growth factor (PGF)-1, accompanied by reduced numbers of peripheral CD8+ T cells, CD4+ T cells, and double-negative T cells [18].

There is little information about the immune response to ZIKV infection in pregnant women. Pregnancy involves complex adaptive and innate immune mechanisms to prevent a maternal response against paternal antigens while retaining competency to protect the mother and fetus against [19,20]. The main immune regulation during this special graft-versus-host situation occurs at the maternal-fetal interface, but systemic immunomodulatory changes also occur as the result of hormonal variations. During pregnancy, immunoglobulin synthesis is increased, whereas the cell-mediated response is decreased. These systemic changes are associated with an altered T helper (Th)1/Th2 balance and a prevailing anti-inflammatory Th2-like profile [21]. However, a Th1/Th17 pro-inflammatory profile appears by the end of pregnancy [22,23].

In order to describe the systemic pro- and anti-inflammatory response during ZIKV infection at different stages of pregnancy, we determined serum levels of cytokines, chemokines, and receptors in serum samples in these kinds of patients and compared them between different groups: healthy nonpregnant women, healthy pregnant women, nonpregnant ZIKV infected women and pregnant ZIKV infected women.

## 2. Material and Methods

### 2.1. Experimental Groups

All samples were obtained from the Central Laboratory of Epidemiology (CLE) of the National Medical Center “La Raza”, Mexican Institute of Social Security (IMSS). A total of 110 serum samples were studied: 96 samples collected between 2016 and 2017 via the epidemiological surveillance system at IMSS, and 14 samples were obtained from healthy nonpregnant women. The samples were selected from the CLE database by means of the following search criteria: sex, reproductive age, endemic area, pregnancy, reverse transcription-quantitative polymerase chain reaction (RT-qPCR) positive to ZIKV. Registers with a RT-qPCR positive for other concomitant arboviruses (DENV) or incomplete data were excluded. Insufficient or contaminated samples were excluded. The samples were divided into five groups. Group one (NPH) comprised 14 samples from healthy nonpregnant women of reproductive age living in Mexico City (a nonendemic area), without a history of travel to endemic areas, who provided informed consent. Group two (NPZ+) included 22 samples from nonpregnant women of reproductive age from endemic areas of Mexico who had confirmed symptomatic ZIKV infection. Group three (PH) was comprised of 30 healthy ZIKV negative pregnant women, whose samples were collected as part of a linear follow-up of the ZIKAlliance protocol [24]. Group four (PWZ+) included 19 samples from pregnant women with confirmed ZIKV infection who were considered as “weakly symptomatic” because they did not meet the operational definition of Zika established in 2016, consisting of a rash with two or more of the following symptoms: fever, headache, edema, petechiae, conjunctivitis, myalgia, arthralgia, pruritus, and retro-orbital pain. Group five (PSZ+) included 25 samples from symptomatic pregnant women who fulfilled the operational definition of Zika and had confirmed ZIKV infection. The combination of Groups four and four are referred to as pregnant women with ZIKV infection (PZ+). Patients with positive results from RT-qPCR tests for DENV or Chikungunya virus (CHIKV) were excluded. Only patients who were ZIKV positive and DENV and CHIKV negative by RT-qPCR were considered for the study.

### 2.2. ZIKV Infection Confirmation

ZIKV infection was confirmed by triplex RT-qPCR VIASURE Zika, Dengue and Chikungunya Real-Time PCR Detection Kit (CerTest BIOTEC, San Mateo de Gallego, Zaragoza, Spain) according to the manufacturer’s instructions. RNA samples were extracted from 140 μL of serum using the QIAmp Viral RNA Extraction Kit (Qiagen, Hilden, Germany). The detection of antibodies against ZIKV and CHIKV in serum was performed using the commercial EUROIMMUN ELISA kit Anti-Zika Virus IgM and IgG, and a EUROIMMUN ELISA kit Anti-Chik Virus IgM (EUROIMMUN, Lübeck, Germany). For detection of dengue antibodies, an IgM and IgG ELISA kit (PanBio Pty Ltd., Brisbane, Australia) was used. All samples were stored at −80 °C.

### 2.3. Determination of Cytokines and Chemokines in Serum

Cytokines were determined using the Human Cytokine Magnetic 25-Plex Panel (Thermo Fisher Scientific, Carlsbad, CA, USA) in a previously validated and calibrated Luminex 200 System (Luminex, Austin, TX, USA), using the software xPONENT V3.1. Analytes included: IL-1β, IL-2, IL-4, IL-5, IL-6, IL-7, IL-10, IL-12, IL-13, IL-15, IL-17A, TNF-α, IFN-α, IFN-γ, GM-CSF, CCL2 (MCP-1), CCL3 (MIP-1α), CCL4 (MIP-1β), CCL5 (RANTES), CCL11 (eotaxin), CXCL8 (IL-8), CXCL9 (MIG), CXCL10 (IP-10), IL-1RA, and IL-2R, according to the manufacturer’s instructions. Standard curves were generated from the reference cytokine gradient concentrations, and the results were expressed in pg/mL.

### 2.4. Determination of Viral Load

ZIKV load was determined using a standard curve by serial dilutions from a pool of ZIKV-positive samples (10^9^–10^0^ copies) by RT-qPCR using the QuantiTect Probe RT-PCR kit (Qiagen, Hilden, Germany) in a 25 μL reaction, with 12.5 μL of 2× reverse transcription master mix, 0.25 μL of QuantiTect RT, 0.25 μL of each primer (1 μM final concentration), 0.15 μL of probe (0.15 μM final concentration), 6.6 μL of water, and 5 μL of RNA (60 ng), as described elsewhere [25]. The Applied Biosystems 7500 Fast system (Applied Biosystems, Foster City, CA, USA) was used with the following protocol: reverse transcription at 50 °C for 30 min, 95 °C for 10 min, 45 cycles of 95 °C for 15 s, and 60 °C for 1 min according to the Mexican national reference laboratory recommendations [26]. The results were reported as copy number/μL.

### 2.5. Statistical Analysis

Heat-map charts of pro- and anti-inflammatory cytokines and chemokines were constructed using the median concentrations. The values were normalized with respect to their respective control: NPH vs. NPZ+ and NPH vs. PH were normalized against NPH, PH vs. PZ+ was normalized against PH. NPZ+ vs. PZ+ was standardized with respect to the relevant controls. NPZ+ and PZ+ by pregnancy trimester were normalized with respect to PH. Statistical analysis was performed with GraphPad Prism 7.0 (GraphPad Software, San Diego, CA, USA). The Mann–Whitney *U* test was used to compare the cytokine concentrations in control and ZIKV-infected groups. The differences were considered significant when *p* < 0.05. The Pearson’s correlation coefficient test was employed to determine the statistical relationship between variables; the association was considered to be present when *r* > 0.3. Tukey’s multiple comparison test was used to compare the number of copies and trimesters of pregnancy.

## 3. Results

### 3.1. General Characteristics of Sample Groups

With the aim of describing the systemic pro- and anti-inflammatory response during ZIKV infection in pregnant women, 25 cytokines, chemokines, and receptors were measured in 110 serum samples of women recollected through an epidemiological surveillance system and divided into five groups as shown in Table 1. Group one (NPH) comprised 14 nonpregnant healthy women with mean age 26.6, ±7.6 years (range 19 to 45); Group two (NPZ+) included 22 samples from nonpregnant ZIKV-positive women with mean age 30.6, ±6.7 years (range 20 to 41); Group three (PH) comprised 30 samples from pregnant healthy ZIKV-negative women with mean age 27.1, ±6.7 (range 17 to 42); Group four (PWZ+) included 19 samples from pregnant women with ZIKV infection who did not meet the operational definition of Zika and were considered weakly symptomatic, mean 30.1 ± 5.6 years (range 20 to 41); and Group five (PSZ+) included 25 samples from pregnant women with confirmed symptomatic ZIKV infection with mean age 26.9, ±3.9 years (range 20 to 37). The mean age for all groups was 27.9 ± 6.3 years (range 17 to 45); the youngest woman belonged to the PH group and the oldest to the NPH group (Table 1).

Among the pregnant women, the PH group included ten (33.3%) pregnant women in the first trimester, 15 (50%) in the second, and five (16.6%) in the third. The PWZ+ group included six (31.6%) pregnant women in the first trimester, 10 (52.6%) in the second, and three (15.8%) in the third. Finally, the PSZ+ group included five (20%) pregnant women in the first trimester, 17 (68%) in the second, and three (12%) in the third trimester; no significant differences were found in the distribution of pregnancy stage between these groups (*p* = 0.7268). The most frequent symptoms for ZIKV-positive groups (NPZ+ and PSZ+) were exanthema (95.5% and 80%), headache (77.3% and 80%), myalgia (77.3% and 68%), pruritus (72.7% and 64%), arthralgia (77.3% and 56%), fever (45.5% and 56%), retro-orbital pain (36.4% and 52%), conjunctivitis (59.1% and 32%), and less frequently edema (9.1% and 4%). There were no significant differences in the proportion of positive symptoms between NPZ+ and PSZ+ groups. Twelve (63.2%) women in the PWZ+ group reported one or more symptoms even though they did not meet the operational definition of ZIKV infection. Among the PH population, one woman (3.3%) reported a history of fever but was asymptomatic at the time of sample collection.

Four samples (18.2%) in the NPZ+ group were positive for ZIKV IgM, and five (22.7%) were positive for ZIKV IgG, whereas six (27.3%) were positive for DENV IgG, but all samples in this group were IgM negative for DENV and CHIKV. Three samples (15.8%) in the PWZ+ group were positive for ZIKV IgG, and three (15.8%) were IgM positive; three samples (15.8%) were positive for DENV IgG, and one (5.3%) was IgM positive, although this sample was RT-qPCR negative for DENV. All samples in this group were also negative for CHIKV. Finally, two (8%) of the samples in the PSZ+ group were positive for ZIKV IgG, and five (20%) for IgM, nine (36%) presented with IgG antibodies to DENV, and all were negative for CHIKV.

### 3.2. Cytokine Profile in Nonpregnant Women with ZIKV Infection

To assess the cytokine profile induced by ZIKV infection, we first analyzed the cytokine serum levels in nonpregnant women infected with ZIKV (NPZ+) in relation to healthy nonpregnant women (NPH). Figure 1 (row A) shows heat-map charts of pro- and anti-inflammatory cytokines and chemokines (the median concentration data are presented in Appendix A). ZIKV infection induced in nonpregnant women significantly higher levels of the anti-inflammatory cytokine IL-10, and increased pro-inflammatory cytokines IL-6, IL-2, IFN-α, and IFN-γ, and decreased levels of IL-1β in relation with uninfected, nonpregnant women. Moreover, ZIKV infection induced an increase in the chemokines CCL3, CXCL9, and CXCL8, while CCL4 and CXCL11 were significantly decreased.

### 3.3. Cytokine Profile in Pregnant Women

Pregnancy is associated with modifications in the immune response that affect the cytokine environment [27]. To address this issue, we next compared the cytokine profile in the serum of not infected pregnant women with not infected nonpregnant women. There were almost no differences between nonpregnant and healthy pregnant women in both pro- and anti-inflammatory cytokines. Figure 1 (row B) shows discrete differences between the groups in both pro- and anti-inflammatory cytokines; IL-1β was decreased in pregnant women (the median concentration values are presented in Appendix A). The main changes were observed with the chemokines CCL3 and CXCL8 that were increased, whereas CCL4, CCL11, and CXCL10 were significantly decreased. Therefore, pregnant women in the absence of ZIKV infection did not show a clear Th1-type regulatory profile, while the chemoattractant chemokines were increased. The analysis by trimester was not able to identify significant differences except for IL-12 and CCL5, which increased in the third trimester.

### 3.4. Cytokines in Pregnant Women Infected with ZIKV

Since pregnancy affects the global response of the mother to microbial infections [21], we next evaluated whether ZIKV infection induced a distinctive serum cytokine profile. First, we analyzed the serum cytokines in pregnant women infected with ZIKV (PSZ+ and PWZ+) compared with that of healthy pregnant women (PH). Figure 1 (row C) shows that pregnant women with ZIKV infection presented a significant increase in IL-10 and pro-inflammatory cytokines IL-6, TNF-α, IL-17A, IFN-α, and IFN-γ but not IL-1β. Although IL-2 seems to be lower in healthy pregnant women, the difference was no significant (Appendix A). Moreover, there were significant increases in serum levels of CCL2, CXCL8, CXCL9, CXCL10, IL-1RA, and IL-2R. However, unlike the changes in the remaining pro-inflammatory cytokines, the decrease in chemokines such as CCL4 and CCL11 was not significant (Appendix A). These data indicated that ZIKV infection during pregnancy modified the inflammatory profile, favoring a regulatory profile by increasing the production of chemokines and of pro- and anti-inflammatory cytokines essential for the antiviral response. Interestingly, IL-1β seems to be downregulated in nonpregnant infected but upregulated in infected pregnant women; however, the difference did not reach statistical significance. Moreover, infected pregnant women showed expression of CCL2, IL-17A, CXCL10, and IL-2R, which were not significant in nonpregnant women.

### 3.5. Cytokine Profiles in Pregnant and Nonpregnant Women Infected with ZIKV

Similar pro- and anti-inflammatory profiles were observed in pregnant (PZ+) and nonpregnant (NPZ+) women infected with ZIKV, except for IL-1β, which was lower in NPZ+ (Figure 1, row A and C). Of note, CCL11 was lower, whereas CCL3 and CXCL8 were higher in NPZ+, and a small increase in CXCL10 was observed in the group of infected pregnant women, although it was not significant (Figure 2A–C and Appendix A). We wanted to know if the viral load was the same; surprisingly, it was higher in pregnant women infected with ZIKV (PZ+) (*p* = 0.0118) (Figure 2, row D). No correlation was found between cytokine concentrations and viral load in pregnant women infected with ZIKV (Appendix A).

### 3.6. Cytokine Profile According to Trimester of Pregnancy

It has been described that cytokine environment changes during the course of pregnancy [27]. Therefore, we evaluated whether the cytokine profile during ZIKV infection differs depending on the trimester of pregnancy. We noticed higher significant values in all anti-inflammatory cytokines (IL-4, IL-5, IL-10, and IL-13) in samples of pregnant women infected with ZIKV from the first trimester of pregnancy and higher CCL3 and CXCL8 in samples from the second trimester of pregnancy; the rest of the cytokines did not reach significant values. (Figure 3 and Appendix A). In contrast, symptomatic pregnant women did not show differences in all cytokines when compared to healthy pregnant women with pregnant women infected with ZIKV (Appendix A).

Since serum cytokines could be related to the clinical features of Zika, we next analyzed the cytokine levels in the serum of symptomatic and weakly symptomatic pregnant women. The differences between the groups were minimal; only IFN-α and CCL11 reached higher significant levels in weakly symptomatic pregnant women infected with ZIKA and CCL3 in symptomatic pregnant women (Figure 4, row A).

On the other hand, we wanted to know whether the cytokines increase affects viral replication and if there is any relationship with the symptoms of the disease. Interestingly, the viral load was higher in weakly symptomatic ZIKV infected pregnant women; however, this difference was not significant (Figure 4, row D).

### 3.7. Cytokine Profile According to Days after Disease Onset

Considering that no difference was found in the cytokine profile between weakly symptomatic and symptomatic pregnant women infected with ZIKA, we further compared the cytokine profile according to the days after the clinical onset of the disease. Figure 5 compares the concentration of each cytokine in pregnant (PZ+) and nonpregnant women (NPZ+) infected with ZIKV according to the number of days after the onset of the disease that the sample was taken. Both anti- and pro- inflammatory cytokines presented higher concentration at the beginning of the disease and a tendency to decrease; however, the correlation values were low, except for IFN-γ that was higher in samples from NPZ+ at day 8. CCL11 was higher in samples from NPZ+. In addition, in the PZ+ group, the concentrations of chemokines and receptors were positively correlated with the number of days after the onset of the disease, except for CCL2, CCL5, and CXCL8. Although the correlation for these chemokines was positive in both groups, there was a greater association in the NPZ+ group and almost none in the PZ+ group.

## 4. Discussion

ZIKV infection during pregnancy generally presents with mild symptoms and is frequently subclinical. However, it is an important public health threat because infection of the fetus may cause growth restriction, central nervous system abnormalities, or even fetal death [28]. In the present study, we measured cytokines in sera from pregnant women with ZIKV infection and compared these with sera from nonpregnant women with ZIKV infection and with pregnant women without infection. The samples from pregnant women positive for ZIKV infection were collected through the IMSS epidemiological surveillance system, and patient records were reviewed to divide the patients into two groups: those who met the operational definition of Zika fever and those who, despite having some clinical manifestations, did not meet this definition, and therefore were considered weakly symptomatic.

It has been considered that a healthy pregnancy correlates with Th2-type immunity, while a Th1-type pro-inflammatory immunity could be harmful for the mother and the fetus [29]. However, the immunological mechanisms that maintain tolerance to the fetus and normal protective responses to pathogens during pregnancy are complex and not clearly understood. Gestation involves the release of neurotransmitters and hormones, such as estradiol and progesterone, largely produced by the placenta and other tissues, and which trigger a complex network of cytokines that drive the maternal systemic tolerance to the developing fetus [27,30] At the beginning of pregnancy, an inflammatory response is elicited that is characterized by circulating cytokines such as transforming growth factor (TGF)-β, CCL2, CCL3, CCL4, CCL5, and prostaglandins. Uterine epithelial cells release other pro-inflammatory cytokines, including GM-CSF [27,31], while the placenta produces IL-18BP, IL-36R, and other cytokines of the IL-1 family [32]. The development and growth of the fetus require remodeling of the uterine spiral arteries, which is in part mediated by TNF-related apoptosis-inducing ligand (TRAIL) [33], while CXCL8 promotes trophoblast invasion and neoangiogenesis by decidual natural killer cells and production of CXCL10, vascular endothelial growth factor (VEGF), and PGF [34]. At the end of pregnancy, the myometrial smooth muscle cell contractions of labor are associated with an increase of pro-inflammatory cytokines such as IL-1β, IL-6, and TNF-α [35], and downregulation of members of the IL-10 family (IL-10, IL-20, IL-22, and IL-28A) [36]. An increase in the concentration of TNF-α, GM-CSF, CCL3, and CXCL8 has been found in the cervix and myometrium during labor [37]. Circulating cytokines in the sera from healthy pregnant women generally show a characteristic Th2-type profile, with increased anti-inflammatory cytokines [38]. However, we did not observe a clear anti-inflammatory type 2 profile in our study, but there was an increase in some chemokines such as CCL3, CXCL8 in normal pregnant women (Figure 1B).

Both anti- and pro-inflammatory cytokines and chemokines were increased in sera from nonpregnant women infected with ZIKV (Figure 1A). It has been reported that Zika fever increases circulating IL-6 levels without changing IL-10 levels [17]. However, it has been shown that a more complex profile is elicited consisting of Th1, Th2, Th9, and Th17 responses during the acute phase of ZIKV infection followed by their decrease in the convalescent phase [17,39]. Sera from convalescent patients were not available for this study because it included only those sent to the Central Laboratory of Epidemiology at the National Medical Center “La Raza” for confirmatory diagnosis.

Viral infections during pregnancy elicit high levels of circulating IFN-γ, TNF-α, IL-6, and TGF-β [40]. This antiviral response is also observed in infections by other flaviviruses (DENV and West Nile virus) [41,42], including acute ZIKV infection in pregnant women [43]. The samples from pregnant women with ZIKV infection included in this study showed increases in all anti-inflammatory cytokines (IL-10), and some pro-inflammatory cytokines (IL-6, IFN-γ, IFN-α, and IL-17A), chemokines (CXCL10, CCL2, CXCL9, and CXCL8), and receptors (IL-1RA and IL-2R) (Figure 1C and Appendix A). The increases in IFN-α and IFN-γ suggest that pregnant women elicit effective antiviral responses, as documented in other studies [8,44]. The high levels of IL-10 suggest an antiviral response that could favor recovery, as reported in mild forms of DENV infection where an increase of IL-10 has been associated with favorable evolution [45,46], while an increase in circulating CCL3, CCL4, CXCL9, and CXCL8 was associated with neurological damage [43,47]. The results reported here are consistent with some studies in patients infected with ZIKV that observed high levels of pro-inflammatory cytokines and other regulatory molecules, including IL-1RA and IL-4, which would suggest that the virus is capable of regulating the inflammatory response by increasing the production of these regulatory cytokines (Figure 1 C) [17,18]. However, other studies have not found differences in the levels of IL-10, IL-4, and IL-6 in pregnant women infected with ZIKV [43].

It has been reported that viremic patients experiencing moderate symptoms show significantly higher quantities of CXCL10, CCL2, IL-1RA, CXCL8, and PGF-1, accompanied by reduced numbers of peripheral CD8+ T cells, CD4+ T cells, and double-negative T cells [18]. However, sera obtained later after the onset of symptoms showed lower concentrations of pro-inflammatory cytokines and higher levels of some cytokines and chemokines associated with T cell activation (Figure 5).

Our results are consistent with the characteristic pattern of circulating CXCL10, CCL2, and CXCL9 chemokines during ZIKV infection [39,43]. The increase in CXCL10 [17], IL-22, CCL2 [43], and TNF-α in pregnant women infected with ZIKV has been associated with fetal malformations [39]. Interestingly, CXCL10 was increased only in samples from pregnant women infected with ZIKV during the first trimester and also in weakly symptomatic pregnant women (Figure 1C, Figure 3 and Figure 4B). The sera from pregnant women infected with ZIKV also showed high levels of CXCL8 (Figure 1C), which is inconsistent with the observations of Foo et al., who found low levels of this chemokine in a similar population [43]. The role of this chemokine during ZIKV infection is unclear, but it is a potent neutrophil activator and promotes human immunodeficiency virus-1 replication in macrophages and primary microglia [48]. During pregnancy, CXCL8 promotes migration of trophoblast cells by increasing matrix metalloproteinases (MMP2 and MMP9) [49]. However, high levels of this cytokine have been associated with miscarriages [50]. Interestingly, the sera from ZIKV-infected women that presented with exanthema showed higher levels of CXCL8 (23.37 vs. 9.95 pg/mL).

Pregnant women with ZIKV infection showed higher viral load than nonpregnant women with infection and weakly symptomatic pregnant women as compared with symptomatic pregnant women (Figure 2D and Figure 4D). Samples obtained early at the beginning of the clinical manifestations showed higher pro- and anti-inflammatory cytokine levels than those taken later after disease onset. However, this trend was not seen with the chemokines because these were higher in the samples obtained long after the clinical onset. This trend was more noticeable in the samples from nonpregnant women. Contrary to previous findings [17], in the present study, the chemokine concentrations observed in pregnant women infected with ZIKV were similar in the samples obtained at all times after the clinical onset of the disease (Figure 5).

Some studies have shown that cross-reactive anti-DENV antibodies can enhance the risk of major clinical manifestations of ZIKV through the mechanism of ADE [51]. Indeed, samples from women with ZIKV infection (NPZ+ and PZ+) who had IgG antibodies to DENV showed higher cytokine levels than those who did not (Appendix A). Although the differences did not reach significance, there is a clear trend, probably caused by cross-reactivity between ZIKV and DENV.

Although the number of studies of the immune response in pregnant women infected with ZIKV and the production of immune mediators has increased significantly since the epidemic behavior of Zika, at the present time, the disease has practically disappeared, and only sporadic cases remain, making the study of the virus study difficult. Therefore, despite the limitations, the sera collected for this study may provide information that contributes to the understanding of the physiopathology of ZIKV infection. The results obtained herein are not sufficient to correlate with neurological damage or congenital malformations in the products. However, the cytokine profile is consistent with other reports. Especially interesting is the observation that weakly symptomatic pregnant women infected with ZIKA presented with a high viral load.

## 5. Conclusions

The data show that nonpregnant and pregnant women infected with ZIKV show the expected broad pro- and anti-inflammatory response with high levels of cytokines and antiviral activity probably mediated by IFN-γ. However, the profile in the weakly symptomatic group did not differ from the group of pregnant women who presented with evident symptoms, which was expected to show a clear pro-inflammatory profile. Moreover, the weakly symptomatic group presented with a higher viral load. The main difference between pregnant women infected with ZIKV was higher CXCL10 and IL-1RA expression in weakly symptomatic pregnant women. The increase of CXCL10 could be related to product damage during pregnancy, especially in asymptomatic patients.

## Figures and Tables

**Figure 1 viruses-13-00080-f001:**
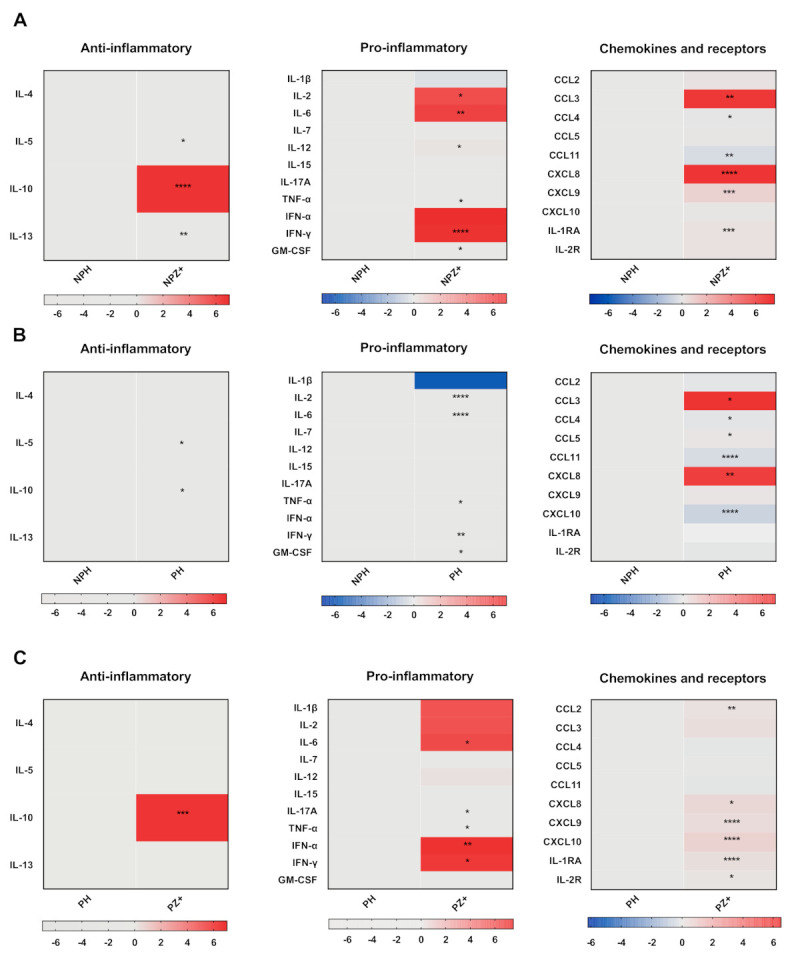
Serum cytokine profile in nonpregnant, pregnant women infected with ZIKV, and nonpregnant healthy women. Cytokines were determined by a Luminex 25-plex assay and grouped into Anti-inflammatory, pro-inflammatory, and chemokines and receptors (columns). (**A**) Nonpregnant healthy women (NPH = 14) vs. Nonpregnant symptomatic ZIKV infected women (NPZ+ = 22), (**B**) NPH vs. healthy pregnant women (PH = 30), (**C**) PH vs. pregnant women with ZIKV infection (PZ+ = 44). The heat map expressed in log10 represents the fold increase concentration in red, decrease in blue, and no change in gray with respect to healthy controls. * *p* < 0.05, ** *p* < 0.005, *** *p* < 0.0005, **** *p* < 0.0001.

**Figure 2 viruses-13-00080-f002:**
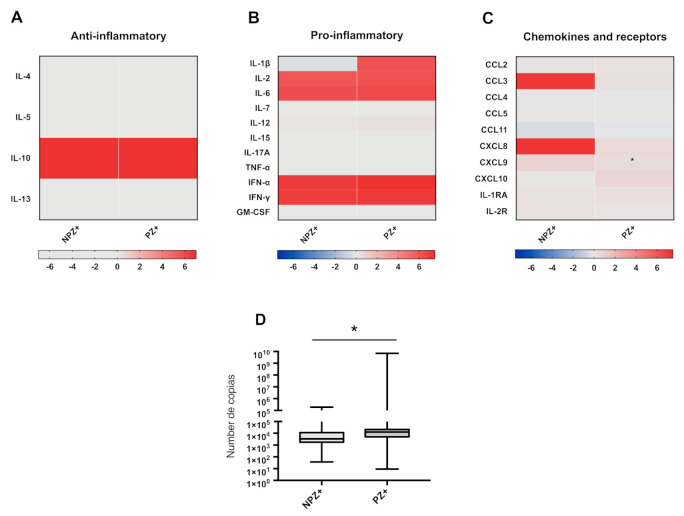
Comparison of the cytokine profile in nonpregnant and pregnant women infected with ZIKV. (**A**) Anti-inflammatory cytokines, (**B**) pro-inflammatory, (**C**) chemokines and receptors. Nonpregnant women infected with ZIKV (NPZ+ = 22) vs. pregnant women infected with ZIKV (PZ+ = 44). The heat map expressed in log10, represents the fold increase concentration in red, decrease in blue, and no change in gray with respect to the healthy control (PH). (**D**) Copy numbers in NPZ+ vs. PZ+ The number of copies was determined by a quantitative RT-qPCR assay and U Mann–Whitney test determined, considering significant difference: * *p* < 0.05.

**Figure 3 viruses-13-00080-f003:**
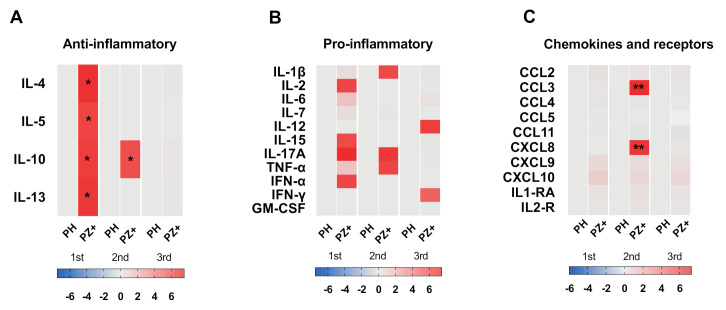
Cytokine profile in pregnant women healthy and infected with ZIKV according to pregnancy trimester. Pregnant women infected with ZIKV (PZ+; 1st = 11, 2nd = 27 and 3rd = 6), healthy pregnant women (PH; 1st = 10, 2nd = 15 and 3rd = 5). Cytokines were grouped into (**A**) Anti-inflammatory, (**B**) Pro-inflammatory, and (**C**) chemokines and receptors. The heat map expressed in log_10_ represents the fold increase concentration in red, decrease in blue, and no change in gray with respect to healthy controls. Tukey’s multiple comparisons test, considering a significant difference: * *p* < 0.05, ** *p* < 0.005.

**Figure 4 viruses-13-00080-f004:**
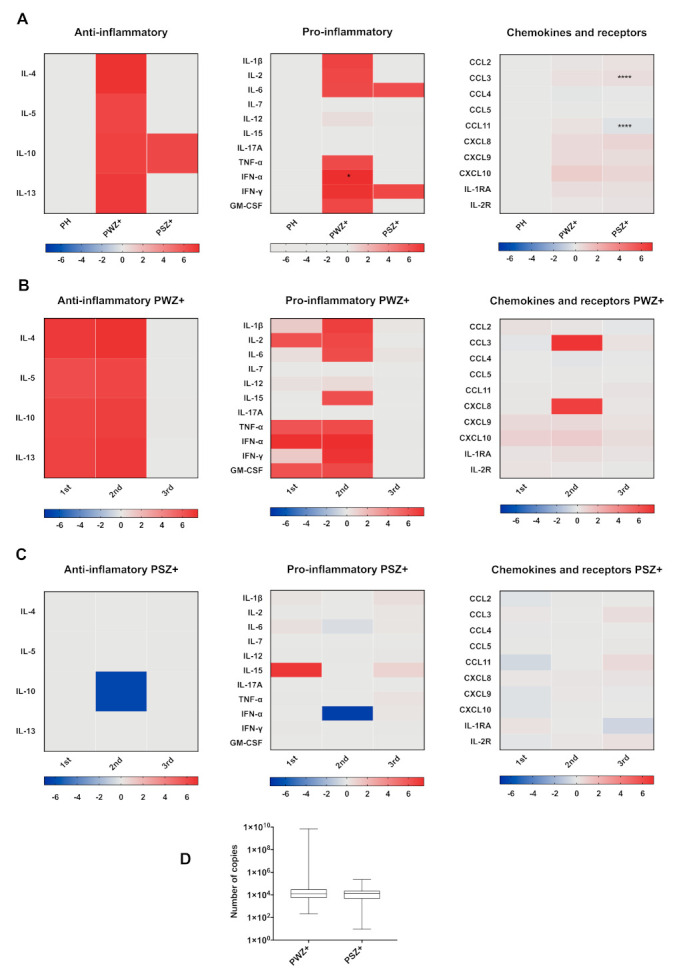
Comparison of cytokine profile in pregnant infected with ZIKV+ with and without symptoms. Cytokines were determined by a Luminex 25-plex assay and grouped into Anti-inflammatory, pro-inflammatory, and chemokines and receptors (columns). (**A**) Healthy pregnant women (PH = 30) vs. weakly symptomatic pregnant women (PWZ+ = 19) and symptomatic pregnant women (PSZ+ = 25). (**B**) PWZ+ by trimester 1st (*n* = 6) vs. 2nd (*n* = 10) and 3rd (*n* = 4). (**C**) PSZ+ by trimester 1st (*n* = 5) vs. 2nd (*n* = 17) and 3rd (*n* = 3). The heat map expressed in log10, represents the fold increase concentration in red, decrease in blue, and no change in gray with respect to the healthy control (PH). (**D**) Viral load (Number of copies) in sera of pregnant women infected with ZIKV, low symptomatic (PWZ+) vs. symptomatic (PSZ+). * *p* < 0.05, **** *p* < 0.0001.

**Figure 5 viruses-13-00080-f005:**
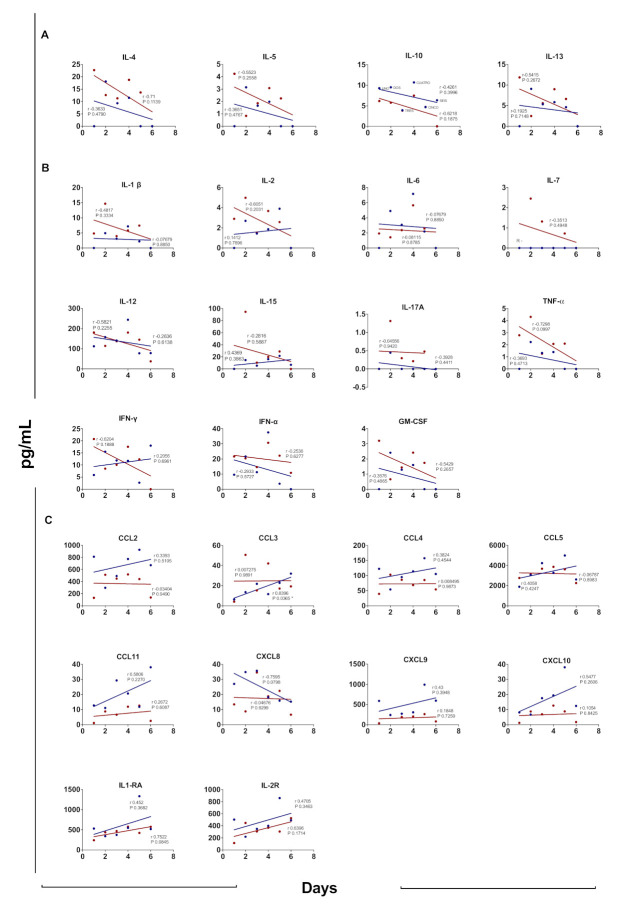
Cytokine concentration by days after disease onset in pregnant and nonpregnant women infected with ZIKV. Serum samples were analyzed in a Luminex assay. Blue lines: Nonpregnant women infected with ZIKV (NPZ+ = 22); Red lines: Pregnant women infected with ZIKV on and (PZ+ = 44) (**A**) Anti-inflammatory cytokines, (**B**) Pro-inflammatory cytokines, and (**C**) chemokines and receptors. Pearson’s test correlation analysis. It was considered a significant difference if *p* < 0.05 and correlation if r > 0.03.

**Table 1 viruses-13-00080-t001:** Epidemiological data of the patients included in the study. Healthy nonpregnant women (NPH); Nonpregnant women ZIKV positive (NPZ+); Healthy pregnant women (PH); Pregnant women ZIKV positive (PZ+); Pregnant women weakly symptomatic (PWZ+); Pregnant women symptomatic ZIKV positive (PSZ+). Not applicable (N/A), not determined (ND).

Clinical Parameters	NPH	NPZ+	PH	PZ+	
				PWZ+	PSZ+
Number of samples	14	22	30	19	25
Age	26.2 (19–45)	30.6 (20–41)	27.1 (17–42)	30.1 (20–41)	26.86 (20–37)
Pregnancy status (weeks)					
First trimester (0–13)	N/A	N/A	10 (33.3%)	6 (31.6%)	5 (20%)
Second trimester (14–26)	N/A	N/A	15 (50%)	10 (52.6%)	17 (68%)
Third trimester (>27)	N/A	N/A	5 (16.6%)	3 (15.8%)	3 (12%)
Days after disease onset (mean)	N/A	3	N/A	3	4
Symptomatology, no. (% total)					
Fever (>38 °C)	0.0	45.5	3.3	63.2	56
Headache	0.0	77.3	6.7	47.4	80
Myalgia	0.0	77.3	6.7	36.8	68
Arthralgia	0.0	77.3	6.7	36.8	56
Retro-orbital pain	0.0	36.4	6.7	36.8	52
Pruritus	0.0	72.7	6.7	68.4	64
Conjunctivitis	0.0	59.1	3.3	52.6	32
Edema	0.0	9.1	0.0	0.0	4
Exanthema	0.0	95.5	6.7	77.3	
Respiratory symptoms or other	0.0	0.0	6.7	0.0	0.0
qRT-PCR (%)					
ZIKV	N/A	100	0	100	100
DENV	N/A	0	0	0	0
CHIKV	N/A	0	0	0	0
Serological tests					
ZIKV IgM	N/D	18.2	0	15.8	20.0
ZIKV IgG	N/D	22.7	0	15.8	8.0
DENV IgM	N/D	-	0	5.3	0.0
DENV IgG	N/D	27.3	0	15.8	36.0
CHIKV IgM	N/D	0	0	0	0.0
Location	Mexico City	Endemic areas of Mexico	Endemic areas of Mexico	Endemic areas of Mexico	Endemic areas of Mexico

## Data Availability

The data presented in this study are available on request from the corresponding author. The data are not publicly available because the database contains personal identification data.

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
