# Peer review of "Pregnant Women Infected with Zika Virus Show Higher Viral Load and Immunoregulatory Cytokines Profile with CXCL10 Increase"

_viruses, 2021, doi:10.3390/v13010080_

Round 1

Reviewer 1 Report

Camacho-Zavala et al. provide an interesting study about the cytokine profile in serum of non-pregnant, pregnant, healthy and ZIKV-positive individuals. Upon streamlining the manuscript, clearly describing the addressed and answered research questions and illustrating the data in a more concise way, this manuscript might give insides into the changes of the immunes system upon infection. Unfortunately in the moment, the manuscript lacks clear structure, so it is hard for the reader to follow the rational of the analyses, and most importantly, to grasp what the main finding is.

General points:

The study reads as superficial as the title: the cytokine profiles are shown, but implications of the findings are unclear. What is the main finding of this article? State this in your title and make it clear in the manuscript.

Please clearly lay out what the research question is and what the results and interpretations are.

The scale in the figures is always different. For comparison reasons it would be better if it was the same throughout.

The main questions are: what is the cytokine profile in:

  • Healthy nonpregnant vs pregnant
  • Healthy nonpregnant vs zikv nonpregnant
  • Healthy pregnant vs zikv pregnant
  • Zikv nonpregnant vs zikv pregnant

It would be best to show one figure including all the data first. Then perform subanalyses according to the research questions. And then performed detailed analysis like severity of disease or trimester.

The discussion needs to be restructured, otherwise it is hard to follow why which analysis was performed, how it fits to previous findings, and what are the new results of this study.

I suggest to structure the discussion as follows (only a suggestion, the crucial part is that there needs to be an obvious theme): How do all the findings compare to (or contradict) previous findings? Have other studies found similar profiles in pregnant women? And in zikv infected non-pregnant? Then what was your question and how was it answered? What is new about the data you provide? Is there something specific about pregnant infected woman that we learn? Then is there something about symptoms differences? Then is there something in trimester? Then what does it mean? What could be the mechanism and what could be the effect on the child or on the virus infection?

Specific:

Line 23. This objective is very vage. Clarify.

Line 36 What is the result? Is infection in pregnant different to non pregnant? And what is the difference and what does it mean? Does this differenece affect ZIKV infection or the child?

line 110: “the samples were selected using the CLE database” – on which basis were they selected? Randomly? Did you exclude some?

Fig 1 scale is not indicated. Figure 1: figure legend does not fit to the figure, since also data from healthy individuals are shown

Fig1C. it jumps out that IL1ß is downregulated in nonpregnant infected but upregulated in infected pregnant. This should be specifically mentioned in the text and an immediate interpretation provided.

Line 228: would make it clearer that there are almost no differences between NPH and PH in both pro- and anti-inflammatory cytokines

Line 231: CCL5 is not blue in figure 1b, but mentioned here

Line 235: why not shown? Can be shown in the supplement

Line 241: text does not fit to shading in figure 1c middle/right

Line 244: why is CCL4, CCL11 dark blue if it not significant?

Fig 2B and C. This data should first be shown as overall pregnant zikv positive vs pregnant zikv negative (if this data is available). Then in further analyses authors can go into subanalyses if this will add information.

Figure 2: subgrouping is an interesting analysis but should be shown later. it should be mentioned that by subgrouping the numbers of samples gets lower, thus also decreasing significance

Fig 2D. Why is the viral load mentioned at this point in the manuscript? What is the rational? What information does it add? Shouldn’t it be mentioned in the beginning as Fig 1? Please explain, why this was analysed and what we are learning from this.

Line 259: is there a difference between the first and third trimester for CXCL8 in the figure 2b?

Line 260: the virus load looks similar. The text says there is a difference. is there significance?

Line 274: on which basis were these three cytokines picked? There are also others with similar differences in figure 3

3.5 please clearly explain the rational why this analysis was done at this time of the study? What is the question behind it and what do we learn?

Figure 4. This illustration is already seen in Fig 1 a and c and should be shown before Fig 2. What is the rational for showing it at this point and not earlier in the manuscript. Wouldn’t it make sense to compare ALL results to each other in Fig 1? If there is one figure showing all data points on the same scale the results will become perfectly comparable at first sight. The next figures then can add information by different comparisons.

Figure 4D: consistent y-axis and English labelling.

Is figure 5 the same as fig 2b or 2c? or combined? Should be made clear. And if so it should all be in one figure and discussed in combination.

Talking about significant up/downregulation (e.g. line 215, 231), would be careful, this needs to be shown by statistics. Some up down regulations seem very minor. Can statistics be included in the heat maps?

Line 294: description of observations in the third trimester are not apparent in figure 5

Why is table 1 again placed after the results part?

Line 327 “It has been considered that a healthy pregnancy correlates with Th2-type immunity, while a Th1- type pro-inflammatory immunity could be harmful for the mother and the fetus.” What is the reference?

Line 333 “the beginning of pregnancy, an inflammatory response is elicited that is characterized by circulating cytokines such as transforming growth factor, CCL2, CCL3, CCL4, CCL5, and prostaglandins.” How was this in your study?

Refer to figures in the discussion, otherwise it is hard to follow if statements are made about literature or about findings in the manuscript

Line 404-408. The fact that something is studied and that this study has results is not a conclusion.

The conclusion is not informative. The fact that ZIKV infection induces inflammatory responses in non pregnant women has been known before. What is new in this study? Aren’t the findings about pregnant infected women more interesting? –Is there a difference between pregnant and non-pregnant women overall or can this not be evaluated?

Line 411 the meaning of this sentence is not clear to me.

Supplementary table 3 is never mentioned in the results part

Author Response

Reviewer 1

  1. The study reads as superficial as the title: the cytokine profiles are shown, but implications of the findings are unclear. What is the main finding of this article? State this in your title and make it clear in the manuscript. Please clearly lay out what the research question is and what the results and interpretations are.

Response 1: The title was modified according to the reviewer's suggestion. The main finding, “Pregnant women infected with Zika virus show higher viral load and immunoregulatory cytokines profile with CXCL10 increase”. Lines 2-4

  1. The scale in the figures is always different. For comparison reasons it would be better if it was the same throughout.

Response 2: The scale was modified and homogenized in all the heat map figures as suggested by the reviewer.

  1. The main questions are: what is the cytokine profile in:
  • Healthy nonpregnant vs pregnant
  • Healthy nonpregnant vs zikv nonpregnant
  • Healthy pregnant vs zikv pregnant
  • Zikv nonpregnant vs zikv pregnant

Response 3. The comparisons are stated as requested by the reviewer at the end of the introduction as follows “In order to describe the systemic pro- and anti-inflammatory response during ZIKV infection at different stages of pregnancy we determined serum levels of cytokines, chemokines, and receptors in serum samples in these kind of patients and compared them between different groups: healthy non-pregnant women, healthy pregnant women, non-pregnant ZIKV infected women and pregnant ZIKV infected women.” Lines 106-110

  1. It would be best to show one figure including all the data first. Then perform sub-analyses according to the research questions. And then performed detailed analysis like severity of disease or trimester.

Response 4. The order of the results was changed only as requested for the trimesters of pregnancy

  1. The discussion needs to be restructured, otherwise it is hard to follow why which analysis was performed, how it fits to previous findings, and what are the new results of this study.

Response 5. The discussion was modified, however not fully restructured

I suggest to structure the discussion as follows (only a suggestion, the crucial part is that there needs to be an obvious theme): How do all the findings compare to (or contradict) previous findings? Have other studies found similar profiles in pregnant women? And in zikv infected non-pregnant? Then what was your question and how was it answered? What is new about the data you provide? Is there something specific about pregnant infected woman that we learn? Then is there something about symptoms differences? Then is there something in trimester? Then what does it mean? What could be the mechanism and what could be the effect on the child or on the virus infection?

Specific:

  1. Line 23. This objective is very vague. Clarify.

Response 6.  The objective was modified as requested to “Objective. To describe the role of pregnancy in the systemic pro- and anti-inflammatory response during symptomatic and low symptomatic ZIKV infection”. Lines 25-26

  1. Line 36 What is the result? Is infection in pregnant different to non pregnant? And what is the difference and what does it mean? Does this difference affect ZIKV infection or the child?

Response 7. The abstract was modified in order to address these questions. The results were clarified “similar profile of inflammatory markers was found in nonpregnant infected women as compared with pregnant infected women, except for CXCL10 that was higher”. Although, it is not possible to answer the question of how this difference affect the child, chemokines such as CXCL10 and CCL2 have been correlated with congenital malformations.  Lines 30-32, 36-41.

  1. Line 110: “the samples were selected using the CLE database” – on which basis were they selected? Randomly? Did you exclude some?

Response 8.  The text was modified to clarify the sample selection as follows “The samples were selected form the CLE database by means of the following search criteria: sex, reproductive age, endemic area, pregnancy, RT-qPCR positive to ZIKV. Registers with a RT-qPCR positive for other concomitant arboviruses (DENV) or incomplete data were excluded. Insufficient or contaminated samples were excluded.” Lines 114-121.

  1. Fig 1 scale is not indicated. Figure 1: figure legend does not fit to the figure, since also data from healthy individuals are shown.

Response 9. The scale is indicated as noticed by the reviewer and the legend was modified as follows. Healthy controls are included for comparison. “Figure 1. Serum cytokine profile in nonpregnant, pregnant women infected with ZIKV and nonpregnant healthy women”. Line 230

  1. it jumps out that IL-1ß is downregulated in nonpregnant infected but upregulated in infected pregnant. This should be specifically mentioned in the text and an immediate interpretation provided.

Response 10. The concern was addressed as follows “Interestingly, IL-1β seems to be downregulated in nonpregnant infected but upregulated in infected pregnant women; however, the difference did not reach statistical significance”. Lines 263-264. Figure 1 (row C) shows that pregnant women with ZIKV infection showed a significant increase in IL-10 and the pro-inflammatory cytokines IL-6, TNF-a, IL-17A, IFN-a, and IFN-g but not for IL-1b (Supplementary Table 3). Lines 254-257.

  1. Line 228: would make it clearer that there are almost no differences between NPH and PH in both pro- and anti-inflammatory cytokines

Response 11: The observation raised by the reviewer was remarqued as follows “There were almost no differences between nonpregnant and healthy pregnant women in both pro- and anti-inflammatory cytokines. Figure 1 (row B) shows discrete differences between the groups in both pro- and anti-inflammatory cytokines, IL-1β was decreased in pregnant women (the median concentration values are presented in Supplementary Table 2). The main changes were observed with the chemokines CCL3 and CXCL8 that were increased, whereas CCL4, CCL11, and CXCL10 were significantly decreased.” Lines 240-245

  1. Line 231: CCL5 is not blue in figure 1b, but mentioned here

Response 12. The observation was interpreted as correctly noticed by the reviewer. CCL5 was deleted from the text.

  1. Line 235: why not shown? Can be shown in the supplement

Response 13: The statement on CCL5 was deleted and shown in the Supplementary Table 4 as not significant.

  1. Line 241: text does not fit to shading in figure 1c middle/right

Response 14: It was clarified as follows: “Figure 1 (row C) shows that pregnant women with ZIKV infection presented significant increase in IL-10 and pro-inflammatory cytokines IL-6, TNF-a, IL-17A, IFN-a, and IFN-g but not IL-1b. Although IL-2 seems to be lower in healthy pregnant women, the difference was no significant (Supplementary Table 3). Lines 254-257

  1. Line 244: why is CCL4, CCL11 dark blue if it not significant?

Response 15: The P value was close to significance (0.19 and 0.06). The concern observed by the reviewer was addressed as follows: “Figure 1 (row C) shows that pregnant women with ZIKV infection presented significant increase in IL-10 and pro-inflammatory cytokines IL-6, TNF-a, IL-17A, IFN-a, and IFN-g but not IL-1b. Although IL-2 seems to be lower in healthy pregnant women, the difference was no significant (Supplementary Table 3). Moreover, there were significant increases in serum levels of CCL2, CXCL8, CXCL9, CXCL10, IL-1RA, and IL-2R. However, unlike the changes in the remaining pro-inflammatory cytokines, the decrease in chemokines such as CCL4 and CCL11 was not significant (Supplementary Table 3)”. Lines 254-260

  1. Fig 2B and C. This data should first be shown as overall pregnant zikv positive vs pregnant zikv negative (if this data is available). Then in further analyses authors can go into sub analyses if this will add information.

Response 16. The figures are presented in the order suggested by the reviewer. First, the comparison between pregnant women infected and not infected with ZIKV.  Figure 2 was replaced and analyzed as suggested by the reviewer. “Figure 2. Comparison of the cytokine profile in nonpregnant and pregnant women infected with ZIKV.” Line 277.

  1. Figure 2: subgrouping is an interesting analysis but should be shown later. it should be mentioned that by subgrouping the numbers of samples gets lower, thus also decreasing significance

Response 17. Figure 2 is presented in the order suggested by the reviewer. The figure was replaced and infected groups (NPZ+ VS PZ+ ) were first analyzed, including their viral load. The group of infected pregnant women per trimester (PZ+) was subsequently analyzed and compared to their PH control. The number of samples per group were added. (A) Healthy pregnant women (PH=30) vs weakly symptomatic pregnant women (PWZ+= 19) and symptomatic pregnant women (PSZ+= 25). (B) PWZ+ by trimester 1st (n=6) vs 2nd (n=10) and 3rd (n=4). (C) PSZ+ by trimester 1st (n=5) vs 2nd (n=17) and 3rd (n=3). Line 277.

  1. Fig 2D. Why is the viral load mentioned at this point in the manuscript? What is the rational? What information does it add? Shouldn’t it be mentioned in the beginning as Fig 1? Please explain, why this was analyzed and what we are learning from this.

Response 18. We decided the following order, first described according to symptomatology, then pregnancy trimester and finally viral load. We did not find any trend accordingly to de symptomatology. However, we found higher viral load in pregnant women infected with ZIKV, especially in the low symptomatic. The text was modified as follows: “On the other hand, we wanted to know whether the cytokines increase affects viral replication and if there is any relationship with the symptoms of the disease. Interestingly, the viral load was higher in weakly symptomatic ZIKV infected pregnant women, however this difference was not significant (Figure 4, row D).” Lines 307-310

  1. Line 259: is there a difference between the first and third trimester for CXCL8 in the figure 2b?

Response 19. There was difference in the second trimester of pregnancy. The question was addressed as follows “The differences between the groups were minimal, only IFN-a, and CCL11 reached higher significant levels in PWZ+ and CCL3 in PSZ+ (Figure 4, row A and Supplementary Table 6). With respect to chemokines, weakly symptomatic pregnant women showed significant increase in CCL3 and CXCL8 only during the second trimester of pregnancy (Figure 4, row B). Lines 307-309

  1. Line 260: the virus load looks similar. The text says there is a difference. is there significance?

Response 20. The observation raised by the reviewer was addressed as follows:  “Interestingly, the viral load was higher in weakly symptomatic ZIKV infected pregnant women, however this difference was not significant (Figure 4, row D).” Lines 309-310

  1. Line 274: on which basis were these three cytokines picked? There are also others with similar differences in figure 3

Response 21. The analysis was modified according to the reviewer´s concern as follows: “Both anti- and pro- inflammatory cytokines presented higher concentration at the beginning of the disease and a tendency to decrease, however the correlation values were low, except for IFN-γ that was higher in samples from NPZ+ at day 8.” Lines 326-328.

  1. 3.5 please clearly explain the rational why this analysis was done at this time of the study? What is the question behind it and what do we learn?

Response 22. The observation was addressed as follows “Considering that no difference was found in the cytokine profile between weakly symptomatic and symptomatic pregnant women infected with ZIKA, we further compared the cytokine profile according to the days after the clinical onset of the disease.” Lines 321-324

  1. Figure 4. This illustration is already seen in Fig 1 a and c and should be shown before Fig 2. What is the rational for showing it at this point and not earlier in the manuscript. Wouldn’t it make sense to compare ALL results to each other in Fig 1? If there is one figure showing all data points on the same scale the results will become perfectly comparable at first sight. The next figures then can add information by different comparisons.

Response 23. The figure was changed as recommended by the reviewer.

  1. Figure 4D: consistent y-axis and English labelling.

Response 24. The y-axis was corrected as noticed by the reviewer.

  1. Is figure 5 the same as fig 2b or 2c? or combined? Should be made clear. And if so it should all be in one figure and discussed in combination.

Response 25: Figure 5 was not the same as Figure 2b or 2c. However, healthy pregnant women and pregnant women infected with ZIKV by trimester, were included in the new Figure 3, as suggested by the reviewer “Figure 3. Cytokine profile in pregnant women healthy and infected with ZIKV according to pregnancy trimester.” The figure is described in section 3.6. Cytokine profile according to trimester of Pregnancy. Line 283 

  1. Talking about significant up/downregulation (e.g. line 215, 231), would be careful, this needs to be shown by statistics. Some up down regulations seem very minor. Can statistics be included in the heat maps?

Response 26. Statistic significant values were marked with * in Figures 1 to 5.

  1. Line 294: description of observations in the third trimester are not apparent in figure 5

Response 27. It was addressed as follows: “It has been described that cytokine environment changes during the course of pregnancy [27]. Therefore, we evaluated whether the cytokine profile during ZIKV infection differs depending on the trimester of pregnancy. We noticed higher significant values in all anti-inflammatory cytokines (IL-4, IL-5, IL-10 and IL-13) in samples of pregnant women infected with ZIKV from first trimester of pregnancy, and higher CCL3 and CXCL8 in samples from the second trimester of pregnancy, the rest of the cytokines did not reach significant values. (Figure 3 and Supplementary Table 5). In contrast, symptomatic pregnant women did not show differences in all cytokines when compared pregnant healthy women with pregnant women infected with ZIKV (Supplementary Table 5).” Lines 285-292

  1. Why is table 1 again placed after the results part?

Response 28. The table was repeated and was deleted.

  1. Line 327 “It has been considered that a healthy pregnancy correlates with Th2-type immunity, while a Th1- type pro-inflammatory immunity could be harmful for the mother and the fetus.” What is the reference

Response 29. The reference is cited as requested by the reviewer with number (29). (Marzi, M.; Vigano, A.; Trabattoni, D.; Villa, M.L.; Salvaggio, A.; Clerici, E.; Clerici, M. Characterization of type 1 and type 2 cytokine production profile in physiologic and pathologic human pregnancy. Clin. Exp. Immunol. 1996.) Line 347.

  1. Line 333 “the beginning of pregnancy, an inflammatory response is elicited that is characterized by circulating cytokines such as transforming growth factor, CCL2, CCL3, CCL4, CCL5, and prostaglandins.” How was this in your study?

Response 30. We did not observe a clear anti-inflammatory type 2 profile.

“Circulating cytokines in sera from healthy pregnant women generally shows a characteristic Th2-type profile of pregnancy [38], with increased anti-inflammatory cytokines. However, we did not observe a clear anti-inflammatory type 2 profile in our study, but there was an increase in some chemokines such as CCL3, CXCL8 in normal pregnant women  (Figure 1B)”. Lines 372-376

  1. Refer to figures in the discussion, otherwise it is hard to follow if statements are made about literature or about findings in the manuscript

Response 31:  Figures are referred in the discussion as suggested by the reviewer.

  1. Line 404-408. The fact that something is studied and that this study has results is not a conclusion.

Response 32. We agree with the reviewer, the paragraph reflects a limitation of the study and was placed before. Lines 441-444

  1. The conclusion is not informative. The fact that ZIKV infection induces inflammatory responses in non pregnant women has been known before. What is new in this study? Aren’t the findings about pregnant infected women more interesting? –Is there a difference between pregnant and non-pregnant women overall or can this not be evaluated?

Response 33:  The conclusion was modified as follows: “The data show that nonpregnant and pregnant women infected with ZIKV show the expected broad pro- and anti-inflammatory response with high levels of cytokines and antiviral activity probably mediated by IFN-g. However, the profile in weakly symptomatic group did not differ from the group of pregnant women who presented with evident symptoms, that was expected to show a clear pro inflammatory profile. Moreover, the weakly symptomatic group presented higher viral load. The main difference between pregnant women infected with ZIKV was higher CXCL10 and IL-1RA expression in weakly symptomatic pregnant women. The increase of CXCL10 could be related to product damage during pregnancy, especially in asymptomatic patients”. Lines 447-454

  1. Line 411 the meaning of this sentence is not clear to me.

Response 34. The sentence was moved and the presence of cross-reactive antibodies to DENV is discussed in lines 43o to 435

  1. Supplementary table 3 is never mentioned in the results part

Response 35. It is mentioned in lines 252, 255, and 382 of the Results section.

Reviewer 2 Report

Manuscript entitled "Profile of immunoregulatory cytokines in pregnant
 women infected with Zika virus" by Authors Elizabeth Camacho-Zavala et al is about change in cytokine levels in the pregnant women infected with Zika virus compared to non-infected pregnant ant non-pregnant women. This study is performed well and the results are presented in a flow. Results are presented in one table and 5 figures. Overall the study is good.

Minor comments:

  1. Is there any institutional ethics committee approval available for this study?
  2. what is the difference between table 1 and table from line # 314?
  3. Authors need to add what was the overall outcome in the fetus caused by change in cytokine levels in the pregnant women infected and non-infected with Zika virus. Is the data available for the comparison? Otherwise, this finding may not add value to the research unless the final outcome indicates about the status of fetus born to infected pregnant mother compared to non-infected pregnant mother. 

Author Response

Reviewer 2

Manuscript entitled "Profile of immunoregulatory cytokines in pregnant women infected with Zika virus" by Authors Elizabeth Camacho-Zavala et al is about change in cytokine levels in the pregnant women infected with Zika virus compared to non-infected pregnant ant non-pregnant women.

This study is performed well and the results are presented in a flow. Results are presented in one table and 5 figures. Overall the study is good.

Minor comments:

  1. Is there any institutional ethics committee approval available for this study?

Response 36. The approval by the National Research Committee at IMSS was included. Also, other information regarding the fellowships was added as follows: “Funding: The protocol was approved by the National Research Committee at IMSS (R-2016-785-076, P.I. VBA) and was supported in part by the Consejo Nacional de Ciencia y Tecnología (CONACYT), grant number FONCICYT 274386 (PI, VBA) and the European Union’s Horizon 2020 research and innovation programme ZIKAlliance under grant agreement No 734548. The funder(s) had no role in study design, data collection and analysis, decision to publish, or preparation of the manuscript.

Acknowledgments: ECZ was recipient of a doctoral fellowship from CONACyT (No. 263525) and IMSS (No. 99097181) for the Departamento de Inmunología. Escuela Nacional de Ciencias Biológicas. Instituto Politécnico Nacional. The authors thank MsSc Julio Alvarado and Dr. Angeles Hernandez-Cueto for their technical support.” Lines 471-479

  1. What is the difference between table 1 and table from line # 314?

Response 37. The table form line 314 was deleted, the information was duplicated.

  1. Authors need to add what was the overall outcome in the fetus caused by change in cytokine levels in the pregnant women infected and non-infected with Zika virus. Is the data available for the comparison? Otherwise, this finding may not add value to the research unless the final outcome indicates about the status of fetus born to infected pregnant mother compared to non-infected pregnant mother.

Response 38.The conclusion was modified as follows: “The data show that nonpregnant and pregnant women infected with ZIKV show the expected broad pro- and anti-inflammatory response with high levels of cytokines and antiviral activity probably mediated by IFN-g. However, the profile in weakly symptomatic group did not differ from the group of pregnant women who presented with evident symptoms, that was expected to show a clear pro inflammatory profile. Moreover, the weakly symptomatic group presented higher viral load. The main difference between pregnant women infected with ZIKV was higher CXCL10 and IL-1RA expression in weakly symptomatic pregnant women. The increase of CXCL10 could be related to product damage during pregnancy, especially in asymptomatic patients.” Lines 447-454. A study following the outcome in the fetus is in progress. However, the data is not available for all the pregnant women.